# Impact of work-family balance results on employee work engagement within the organization: The case of Slovenia

**Jasmina Žnidaršič** *, **Mojca Bernik**

Faculty of Organizational Sciences, University of Maribor, Kranj, Slovenia

* jasmina.znidarsic@um.si

**Citation:** Žnidaršič J, Bernik M (2021) Impact of work-family balance results on employee work engagement within the organization: The case of Slovenia. PLoS ONE 16(1): e0245078. https://doi.org/10.1371/journal.pone.0245078

**Data Availability Statement:** All relevant data are within the manuscript and its Supporting Information files.

**Funding:** The author(s) received no specific funding for this work.

## Abstract

### Background and purpose

Organizations strive to increase the work engagement of their employees, as engaged employees are more productive employees, but often neglect the significant effects of work-family balance on work engagement. Numerous studies confirm the importance of work-family balance and work engagement, but there is lack of research that explores the relationship between the concepts. Our research fills a research gap in investigating the impact of work-family balance on work engagement, both directly and through individual perceptions of organizational support for work-family balance. The main aim of our research is to empirically test the relationships between the policies and practices of organizations regarding work-family balance, work-life balance and work engagement.

### Methods

Using validated questionnaires, we collected data on organizational support for work-family balance (family-friendly policies and practices, support by leader, support by co-workers, working hours and complexity of work), work-family balance and work engagement. The quantitative data for our analysis was collected through a survey of 343 online participants who were employees in various positions in companies in Slovenia.

### The results

Our results show that the organization's work-family balance policies and practices, such as support by leader, co-workers, and family-friendly policies and practices, have a positive impact on the individual's work-family balance, that work-family balance leads to an increase in work engagement, and that the individual's perception of the organization's work-family balance support leads to an increase in work engagement.

### Conclusion

Knowledge of important work-family balance implications with an understanding of organizational support for work-family balance and the relationships between the constructs of work-family balance and work engagement can be beneficial to business leaders. This

**Competing interests:** The authors have declared that no competing interests exist.

understanding can help them to strengthen employee work engagement through family-friendly policies and practices, and thereby contributing to the area of employee behavior and improving employee productivity.

## 1 Introduction

In today's fast-paced life, individuals often face the problem of how to balance all roles and responsibilities in life, especially those related to work and family. The extent of the conflicts between work and family is related to the increasing participation of women in the work process and the "modernisation" of life [1]. However, Stier et al. [2] argue that the conflict between work and family is not only a reflection of changes in roles according to gender or economic activity of women, but that the relationship is more complex and takes place in an institutional context. Work-family balance is now almost no longer just an individual problem, but is already becoming a social and institutional problem facing all countries. In theory, work-family balance is gaining in importance, but practice shows that there are still difficulties in exercising certain rights related to work-family balance and that organizations still do not recognize the importance of reconciling work and family life [3].

There are two terms in the literature to define a concept that describes the search for a balance between work and other life roles, namely work-life balance and work-family balance [4]. Work-life balance is a concept that supports the efforts of individuals to divide their time and energy between work and other important roles and responsibilities in their lives, such as family, friends, community, spirituality, personal growth, hobbies, and other personal activities [5]. On the other hand, work-family balance is the fulfillment of role-related expectations negotiated and shared between an individual and his or her role-related partners in work and family [6]. The terms are similar, but not identical, as the construct of work-life balance is more comprehensive and broader than the construct of work-family balance [4]. In this paper we use the term work-family balance, because we are only interested in the relationship between work and family.

If we summarize the individual definitions of work-family balance, the concept can be described as satisfactory inclusion or adjustment between two roles in an individual's life, namely work and family [7]. The concept of work-family balance is mainly used to describe the stability and balance between responsibilities related to work and family. Balance being defined by what the individual believes is right. [8]. Work-family balance is therefore understood as the effort of an individual to have enough time and energy to devote to the family, while performing all the tasks in his or her workplace.

In recent years, much research has been conducted in the field of work-family balance. It has been shown that the organization which creates favorable family-friendly work conditions plays an important role in the work-family balance of employees [9,10]. Individuals want to be valued and respected by the organization as employees and as individuals with a private life [11]. One study [12] also confirm the importance of the individual's perception that the organization helps him or her to balance work and family, no matter how well he or she actually does.

The organization can support employees in work-family balance with various family-friendly policies and practices. The most common ones mentioned in the literature are flexible working hours, part-time work, "compressed" work week, flexible arrival times, working from home, holidays [13–15], co-worker support, organizational culture in general [16], and leader support [17].

A study by Grover and Crooker [12] showed that employees are more committed to an organization that offers family-friendly measures, regardless of how much the individual personally benefits from these measures. The relationship of employees to the organization is therefore only better if there is a policy and practice of work-family balance, regardless of whether they themselves benefit from it or not. The perception of organizational support for employees reduces conflicts between work and family [11] and also has an impact on personal, family, and professional outcomes [18].

Research confirms that work-family balance is important both for the individual and for the organization, as it affects job satisfaction, organizational commitment, productivity, performance, efficiency, and retention of existing employees [10]. It also serves as a mediator between work-family conflicts and satisfaction [19]. The literature in the field of work-family balance [20,21] emphasizes the importance of work-family balance and reducing conflicts between the two for health and overall well-being. If perception of an individual is that the organization is family-friendly is also important to reduce the conflict between work and family [11].

A study by Bandekar and Krishna [22] shows that work-family balance practices create a "win-win" situation for both employees and the organization. Organizations that support work-family balance policies and practices benefit in a number of areas, such as reduced staff turnover and sick leave, as well as increased productivity, motivation, job satisfaction, and commitment [23]. The key question for organizations should therefore be how to prevent conflicts between work and other roles such as family [24].

In addition to the problem of work-family balance, organizations are also confronted with the problem of a lack of engagement in working life. Data from a Gallup survey of 155 countries for the year 2016 show that worldwide only 15% of employees were engaged at work, with 67% being unengaged, and 18% actively unengaged. In comparison, the best companies in the world have around 70% engaged employees [25]. Organizations today need employees who are energetic and committed at work, or in other words, organizations need engaged employees [26], because engaged employees are more productive employees [27].

Work engagement is defined as a positive work-related state characterized by vigor, dedication, and absorption [28,29]. Vigor means that the individual has a high level of energy during work and is mentally resilient. Dedication refers to the fact that the employee is strongly involved in their work and at the same time experiences a sense of importance, enthusiasm and challenge. Absorption means that the individual is completely immersed in his or her work with minimal mistakes. [28,30]. Work engagement is therefore a relatively permanent state of mind that refers to the simultaneous investment of personal energy in work experience or success [31].

The concept of work engagement is the subject of many studies. In the literature, work engagement is characterized as an important outcome of a healthy work environment, so it is important, if not necessary, that organizations work with a sense of what their employees need in the work environment. In this way, they stimulate work engagement and thus higher productivity [32]. Work engagement involves the emotional and psychological relationship between employees and their organization, which can be transformed into negative or positive behavior that employees display in the workplace [33,34]. As Taghipour [35] also says, a person is generally engaged when they feel valued and included.

Research shows that work engagement has a number of positive effects that are important both for the work organization and for the individual. One study [36] summarizes the effects of work engagement that were investigated in three groups: performance, professional results, and personal outcome. Work engagement therefore has a significant positive relationship with work outcomes [37] and with the outcomes of employees beyond work [38].

A successful work-family balance also has an impact on the work engagement of individuals [7]. The perception of work-family balance is related to the individual's sense that the organization supports him or her and provides some value to the organization [32]. Another reason is that those employees who are responsible for caring for their families in addition to their work constantly suffer from a lack of time and energy. In these cases, excessive work engagement can turn into burnout, which in turn means less work engagement in the long term [39]. A stronger link between negative family influences on work engagement was observed among women, as research on engagement in different roles [40] showed that women have more links between work and family than men.

There is a lack of research that would examine the direct effects of work-family balance on work engagement or the effects of the organization's support for work-family balance on work engagement, so we decided to fill this research gap in this paper. Research that would investigate the direct impact of an individual's work-family balance on their work engagement was not found, but there are some related previous studies that have significantly influenced the formulation of following hypothesis. For example, the results of a study by Kar and Misra [41] confirmed that those employees who are supported by the employer in balancing work and family life are more satisfied and commited in the workplace. At the same time, research [35,42,43] shows that family-friendly measures taken by the organization influence the greater engagement of individuals in the workplace.

## 1.1 Hypotheses

On the basis of the above, we have formulated three hypotheses to determine the relationship between organizational support for work-family balance, work-family balance, and job engagement. We propose that organizational support for work-family balance has a positive effect on the individual's work-family balance, that work-family balance has a positive effect on the individual's work engagement, and that the perception of organizational support for work-family balance has a positive effect on the individual's work engagement.

Numerous studies underline the importance of support in promoting a working environment that helps employees to balance work and family life [9,44,45]. Most research emphasizes that organizations can support the work-family balance of their employees through family-friendly policies and practices that help employees balance work and family responsibilities [46]. In any case, as Knaflič et al. [23] emphasize, it is important that organizations implement a sufficient number of policies and that these are adapted to the organization and the needs of its employees. It can also be emphasized the important role of work-family balance, and communication technologies and measures that ensure smooth and safe work from home. As research shows [e.g. 47,48] the family-friendly work environment and safety concerns are an important factor in work engagement. Research [2,49,50] has also shown that the length of working time also plays an important role in reconciling work and family life, as those employees who have a longer working day have greater difficulties in work-life balance. Research [50] has shown that in addition to the length of the working day, the complexity of the work also has a negative influence on the work-family balance. In particular, a leader has influence on the creation of a working environment that supports the work-family balance and as one study has shown [23], leader support is also the key to the successful implementation of family-friendly measures. Leader support is important for the work-family balance in all career phases, especially at the beginning of the career and before retirement [51]. Study by Griggs et al. [52] has shown that, in addition to the leader, co-worker support also makes an important contribution to a stimulating work environment and the work-family balance. Based on previous research that emphasizes the importance of elements at the organizational level, such

as the leader support, co-workers support, family-friendly policies and practices, as well as the length of the working day and the complexity of the work, we propose hypothesis 1.

**H1:** Elements at the organizational level influence the work-family balance of the individual.

A study examining the direct impact of the work-family balance of an individual on his or her work engagement has not yet been conducted, but the results of a survey [41] confirmed that those employees who receive employer support in coordinating work and family are more satisfied at work and more engaged. At the same time, research has also shown that family-friendly organizational measures influence an individual's greater work engagement [35,42,43]. Based on this previous knowledge we formulated Hypothesis 2.

**H2:** Work-family balance influences the work engagement.

Research [18] confirmed that job requirements and family-friendly policies and practices shape individuals' perceptions of organizational support in work-family balance. This happens through two mechanisms: on the one hand, the organization shows that it cares about the balance or work-life balance of employees and, on the other hand, it helps them to develop and maintain ways to meet the needs and demands of work and daily life. A study by Swanberg et al. [53] found that some family-friendly measures, such as supporting managers and giving employees control over their schedules, not only help to balance work and family, but also have a significant impact on increasing employee engagement. Work engagement is also influenced by support of work organization, [42] co-workers, and the social environment [43] or, as defined by Taghipour and Dezfuli [35] the moral climate in the organization. Based on this previous knowledge we formulated Hypothesis 3.

**H3**: Individuals' perception of the organization's support for work-family balance influences their work engagement.

The proposed hypotheses were tested in the proposed model (Fig 1) as follows.

## 2 Methods

In the following, measurement instruments and data collection for the purpose of testing hypotheses will be presented, as well as a description of the sample.

### 2.1 Instruments

The hypotheses presented were tested with quantitative methods. This means that primary data were obtained using a survey questionnaire, which were then processed in the SPSS

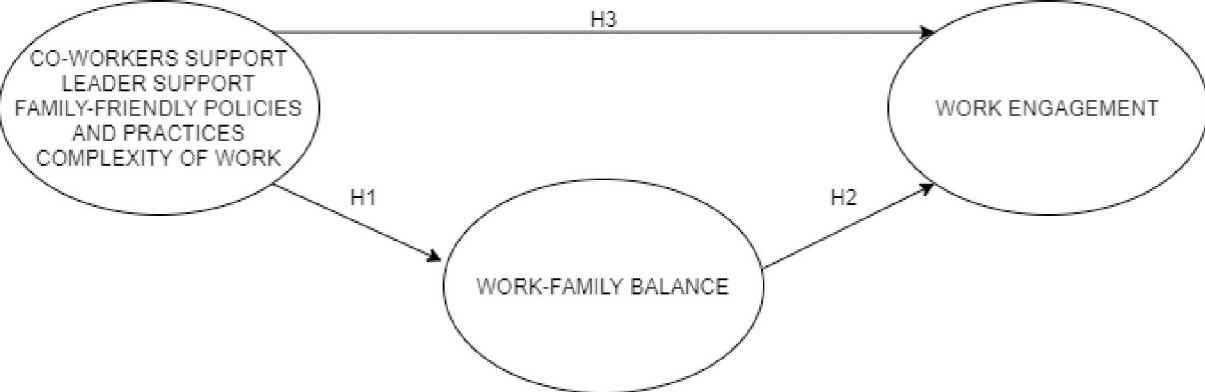

**Fig 1. Model of the connection between the elements within the organization, work-family balance and work engagement.**

program using various statistical methods. With the exception of demographic variables, we used previously used and validated questionnaires, which were adapted accordingly.

The first part of the questionnaire (general part) consists of basic questions about the respondent, such as gender, age, education, size of enterprise, number of working hours per week, legal status, number of children under 18.

The next part of the questionnaire is the questions that relate to the organization's support for the employee's work-family balance. The first two sets of questions relate to the respondent's opinion on how the leader's and the co-workers' support work-family balance. Support from the leader was measured using a nine-item questionnaire developed by Shinn et al. [54]. Respondents were asked to rate on a five-point Likert scale from 1 (never) to 5 (very often) for how frequent an individual leader uses certain supportive behaviors to balance an employee's work and family life. Questions such as working overtime, taking leave to adjust family responsibilities, or listening to employee problems were asked. The reliability coefficient (Cronbach's Alpha) was 0.851. This questionnaire was then adjusted accordingly to obtain an opinion on the support provided by colleagues. The reliability coefficient (Cronbach's Alpha) for this questionnaire was 0.909. The last group of questions in this section related to the organization's practices for balancing work and family life. For this section we used a set of the most frequently used measures in Slovenian companies within the Family-Friendly Company certificate [55]. This questionnaire contains 25 variables divided into 8 sets: working time, work organization, workplace (workplace flexibility), information and communication policy, leadership skills (strategy/management philosophy), human resources development, salary structure and reward benefits, and services for families. Respondents rated each statement using the five-point Likert scale, where 1 meant that they completely disagreed and 5 meant that they strongly agreed that the organization offers them a particular measure. The reliability coefficient (Cronbach's Alpha) was 0.923.

The next set of questions measured the work-family balance of the respondent. Work-family balance was measured by using a four-item scale developed by Brough et al. [56]. The respondents used a five-point Likert scale ranging from 1 (completely disagree) to 5 (completely agree). Respondents rated the statements regarding their work-family balance such as »*The current relationship between the time I spend on work and the time I have for family activities seems good to me*« or »*Overall, I think my work and family life is balanced*«. The reliability coefficient (Cronbach's alpha) was 0.869.

The last set of questions measures the construct of work engagement. To determine the individual's work engagement, we used the Utrecht work engagement scale [30]. The shorter version consisting of nine statements, three for each dimension of engagement were used. According to the Utrecht work engagement scale, engagement consists of three parts: vigor, dedication, and absorption. Respondents indicated on a 5-point scale (1—I do not agree at all, 5—I completely agree) the extent to which they agree with the statements describing their feeling in the workplace (e. g. »*I am full of energy at work*« or »*I am happy when I work intensively*«). The coefficient of reliability (Cronbach's alpha) was 0.865.

To obtain data, we designed an online questionnaire which was sent by e-mail in autumn 2019. The survey was conducted in the form of a cross-sectional design. A quota sample was formed according to the size of the enterprise with excluding micro-enterprises due to the large number of enterprises. According to the Statistical Office of the Republic of Slovenia, in 2017 (the last available data in 2019, when the survey was conducted), 195,756 companies were registered in Slovenia, of which 185,997 micro-enterprises (0 to 9 employees) and 7,329 small enterprises 49 employees), 2,084 medium-sized enterprises (from 50 to 249 employees) and 346 large enterprises (over 250 employees) [57]. The quota sample covered about 0.5% of Slovenian enterprises, i.e. 2 large enterprises, 10 medium-sized enterprises and 37 small

enterprises. Within the sample, we limited ourselves to the private sector, and in the case of large and medium-sized enterprises, we concentrated only on manufacturing enterprises, since in this case we could cover both administrative and production staff. As we are interested in the perception of all employees, the survey covered employees at all levels, from the highest to the lowest, i.e. leaders and all other employees. In the Republic of Slovenia there are current 968,000 persons in employment [58]. A total of 3,200 employees were in the sample. We received 343 fully completed surveys back. We used IBM SPSS Statistics 24 to process and analyse the data.

## 2.2 Participants

The sample size was 343 respondents from different enterprises in Slovenia, of which 48.1% were women and 51.9% men. The age of the respondents ranged from 19 to 62 years with an average age of 39.7 years. By marital status, the structure of the sample was as follows 17.5% were single, 75.5% were married, 4.4% were divorced, and 2.6% were widowed. By educational level, 9.3% of the respondents had completed primary school, 47.8% had completed secondary school, 34.7% had completed higher or university education, 7.6% had a master's degree, and 0.6% had a doctorate. From the data we can see that most of the respondents (82.5%) had completed secondary school or higher education. Of the 343 respondents, 72.3% said they had children, 27.4% said they had no children, and 1 answer was not valid. On average, respondents worked 40.82 hours per week (standard deviation 5.088), and most (80%) 40 hours per week. The minimum was 8 hours per week and the maximum 80 hours per week. 53.4% of the respondents are employed in a small organization, 23% in a medium-sized organization, and 23.6% in a large organization. None of the respondents consider their work very uncomplex, a very small percentage of respondents (2.9%) considered their work uncomplex and 27.1% moderately complex. 49% of respondents, or almost half of them, considered their work complex, and 21% considered it very complex.

In the research, positions or jobs were divided into three groups, i.e. management, administration, and production. Each of the three groups has its own specifics important for the work-family balance (eg responsibility, the possibility of flexible place and time of work, the complexity of work, etc). In our case, management means lower, middle, and top management or leaders whose task is management and leadership; administration means all support and operational services that are important to the functioning of the organization and are not management or production (for example, finance, marketing, human resources and others). 47.5% of the respondents were employed in production 32.4% in administration and 20.1% in management positions.

In the following we combined the results of the previous research, compared them with the results of our own research, and built up new knowledge about the difference. Based on a thorough analysis of the obtained results, we first tested the hypotheses put forward using various statistical methods, which are presented below.

## 2.3 Basic analises for variables

Mean values and standard deviations were calculated for individual questionnaires in relation to the variables included in the study. We were interested in how the sample generally assessed the variables under investigation. The results are presented in Table 1.

Respondents rated the complexity of their work on a 5-point scale from 1 (not complex at all) to 5 (very complex). As can be seen from Table 1, on average, respondents generally rated their work as complex. The average number of working hours is 40 hours per week as most respondents answered that they worked the usual 40 hours per week. Respondents were also

**Table 1.** Mean values, standard deviation and median for dependant and independant variables.

| Variable | Mean | SD | Median |
|---|---|---|---|
| Work complexity | 3.88 | .765 | 4.00 |
| Working hours/week | 40.8 | 5.088 | 40.0 |
| Leader support | 3.50 | .828 | 3.56 |
| Co-worker support | 3.68 | .886 | 3.67 |
| Family friendly policies and practices | 2.74 | .810 | 2.74 |
| Work-family balance | 3.18 | 1.093 | 3.25 |
| Work engagement (overall) | 3.46 | .730 | 3.56 |
| Vigor | 3.52 | .717 | 3.67 |
| Dedication | 3.44 | .910 | 3.67 |
| Absorption | 3.42 | .903 | 3.67 |

asked to assess how often an individual leader and his or her co-workers use certain supportive behaviors to balance an employee's work and family life (1 meant never and 5 meant always). In general, respondents rated both leader and co-workers' support as relatively high. The average co-worker support overall was rated slightly higher than leader support and the standard deviation was higher.

In connection with the organization's family-friendly policies and practices, we asked the respondents what measures their organization implements. Most prevelant are measures in the area of work flexibility (e.g. flexible work breaks, flexible working hours, flexible arrival and departure times for work) and measures for self-control or independence (e.g. independence in planning annual leave, organizing substitutions, and on-call times). Least prevelant are measures related to the protection of workers' children (e.g. various forms of day care for workers, organized holiday care for workers, and the possibility of bringing their children to work part-time). Overall, respondents rated the representation of measures in their organization as relatively poor, averaging 2.74 (1 means "strongly disagree", 5 "very agree").

Averages and standard deviations for the dependent variables were also calculated. The values for work-family balance are slightly above average, which means that the respondents are more satisfied than dissatisfied with work-family balance, but the the standard deviation is quite high (SD = 1.1). The respondents also rated their work engagement relatively high, with the dimension of vigor being highest, and that of absorption lowest.

The sample consisted of 178 men and 165 women. Gender plays an important role in work-family balance. Data shows that the sample was balanced by gender. This is important so that we could look at how both groups saw individual independent and dependent variables, or whether there are significant gender differences in the valuation of individual variables. To examine whether there are statistically significant differences between the gender in the perception of elements of organization, work-family balance, and work engagement, a T-test was used. The results of the T-test showed statistically significant differences between the gender only for the variables of work complexity and work-engagement ($p < 0.5$). Men rated their work as more complex than women, as the average number of responses on the complexity of their work was 4.01 (men) and for women 3.65 (1 = very uncomplex, 5 = very complex). In terms of work engagement, the answers showed that women are statistically significantly more engaged at work than men. The average for men was 3.38 and for women 3.54.

For the other variables, the differences by gender were not statistically significant, but the values still differed slightly between them. Men worked an average of 40.9 hours per week and women 40.8 hours. Leaders' support was rated almost equally by both sexes, with an average of 3.51 for men and 3.50 for women. There was a slightly larger difference in the support of co-

workers, where the average for men was 3.61 and for women 3.74, which means that women rated their co-workers' support in work-family balance slightly better. Women also rated the presence of family-friendly policies and practices in their organization better, as the average for women was 3.03 and the average for men was 2.92. There were also minor gender differences in the assessment of work-family balance performance. Men rated work-family balance somewhat surprisingly worse than women, with the average being 3.14 for men and 3.23 for women.

## 3 Results

This section will discuss the results of analyses for the set hypotheses to determine the relationship between organizational support for work-family balance, work-family balance, and job engagement.

### 3.1 Hypothesis 1—Elements at the organizational level influence the work-family balance of the individual

In the first hypothesis (H1) we determined whether factors at the organizational level influence the work-family balance. The elements within the organization whose effects we examined were working hours, the complexity of the work, the leader support, the support of co-workers, and family-friendly policies and practices. The impact of the elements at the level of the organization on the work-family balance was calculated by a regression analysis (Table 1).

The complexity of the work was rated from 1 to 5, where 1 means that the work is very uncomplex and 5 means that the work is very complex. The complexity of the work was an independent variable and the work-family balance depended on it. The results of the regression analysis showed that the complexity of work had a statistically significant negative impact on the work-family balance of the individual ($p < 0.05$). The less complex the individual's work is, the easier it is to balance work and family, or vice versa, the more complex the work, the more difficult it is to balance work and family. It can be confirmed that the factor "complexity of work" within the factors at the level of the organization influences the work-family balance. The regression analysis also showed a significant influence of family-friendly policies and practices on work-family balance ($p < 0.05$). This means that the more family-friendly policies and practices are available to employees, the easier it will be to balance work and family life. In contrast to the two previous variables, the results of the analysis showed that the number of hours worked per week did not have a statistically significant impact on the work-family balance of a person ($p > 0.05$). We conclude that the results are such, since in most cases (up to 81%), respondents answered that they worked a normal 40-hour week and the ability to balance work and family differed between them (Table 2).

The following two elements, namely the impact of leader support and the impact of co-workers support on work-family balance, were also examined by means of a regression

**Table 2. Estimation of regression coefficients of work complexity, number of working hours per week, and family-friendly policies and practices, and work-family balance.**

| Model | Unstandardized Coefficients | | Standardized Coefficients | Adjusted $R^2$ | t | Sig. |
|---|---|---|---|---|---|---|
| | Beta | Std. Error | Beta | | | |
| Constant | 2.676 | .546 | | | 2.939 | .000 |
| Work complexity | -.165 | .071 | -,115 | 0,158 | 2.392 | .021 |
| Working hours/week | -.007 | .011 | -.033 | | -.605 | .508 |
| Family friendly measures | .523 | ,067 | .387 | | 8.544 | .000 |

**Table 3. Estimation of regression coefficients of the leader and co-workers support and work-family balance.**

| Model | Unstandardized Coefficients | | Standardized Coefficients | Adjusted R$^2$ | T | Sig. |
|---|---|---|---|---|---|---|
| | Beta | Std. Error | Beta | | | |
| Constant | 1.454 | 0.268 | | | 5.417 | 0.000 |
| Leader support | 0.261 | 0.084 | 0.198 | 0.108 | 3.122 | 0.002 |
| Co-worker support | 0.221 | 0.078 | 0.179 | | 2.830 | 0.005 |

analysis. The results of the analysis (Table 3) showed that both factors within the organization examined have a statistically significant influence on the individual's work-family balance ($p < 0.05$). We can conclude from the results that those individuals who receive more support from their leader and co-workers in work-family balance will find it easier to balance work and family life.

As the results of the analyzes show, all the factors examined have a statistically significant influence on the balance of work and family life at organizational level (complexity of work, leader support, co-workers support, and family-friendly policies and practices), with the exception of working hours. Hypothesis H1 can be confirmed and we can say that factors at the level of the organization have a statistically significant influence on the work-family balance of an individual.

## 3.2 Hypothesis 2—Work-family balance influence the work engagement

In the second hypothesis (H2), we determined whether the work-family balance has an impact on work engagement. The influence of work-family balance on the work engagement of employees was verified by a regression analysis. The results of the analysis (Table 4) showed that the work-family balance had a statistically significant influence on the work engagement of individuals ($p < 0.05$). The influence is strong and the explanation for the variable work engagement through work-family balance is also significant. Thus, those individuals who are better able to balance work and family life are more engaged at work.

## 3.3 Hypothesis 3—Individuals' perception of the organization's support for work-family balance influences their work engagement

In order to confirm the last hypothesis (H3), which emphasizes whether the employee's perception of the organization's support for balancing work and family life influences his or her work engagement, we have calculated regression analyzes. We examined the effects of three elements of the organization, namely leader support, co-worker support, and family-friendly policies and practices, which were independent variables, while work-family balance was a dependent variable.

The results of the regression analysis showed that the employee's perception of the leader and co-workers in balancing work and family life had a statistically significant influence on the individual's work-family balance ($p < 0.05$). The more the individual feels that the leader and co-workers support him or her in balancing work and family life, the more engaged he or she

**Table 4. Regression coefficient of the impact of work-family balance on employee work engagement.**

| Model | | Unstandardized Coefficients | | Standardized Coefficients | Adjusted R Square | T | Sig. |
|---|---|---|---|---|---|---|---|
| | | B | Std. Error | Beta | | | |
| | Constant | 2,504 | ,109 | | | 23,037 | ,000 |
| | WLB | ,299 | ,032 | ,448 | ,198 | 9,255 | ,000 |

**Table 5. Regression coefficients of the impact of the leader and co-workers support in work-family balance and the family-friendly policies and practices, on work engagement.**

| Model | | Unstandardized Coefficients | | Standardized Coefficients | Adjusted $R^2$ | t | Sig. |
|---|---|---|---|---|---|---|---|
| | | B | Std. Error | Beta | | | |
| | Constant | 1.855 | .178 | | .198 | 10.399 | .000 |
| | Leader support | .126 | .055 | .144 | | 2.294 | .022 |
| | Co-worker support | .146 | .050 | .177 | | 2.923 | .004 |
| | Family friendly measures | .226 | .048 | .251 | | 4.681 | .000 |

is to the work. Similarly, the results of the regression analysis also confirmed the positive effects of family-friendly policies and practices on work engagement (p <0.05). The influence of the elements studied on work engagement is strong, with the support of leader, co-worker, and family-friendly policies and practices we can explain 20% of the variance in work engagement (Table 5).

On the basis of the results of the regression analysis, which confirmed the statistically significant positive effect of leader support and co-workers support in balancing work and family life and of family-friendly policies and practices on individual work engagement, we can say that the perception of organizational support by individuals in balancing work and family life has a positive influence on work engagement. This confirms hypothesis 3.

On the basis of the confirmed hypotheses, we can set up a model (Fig 1). The model shows that the elements at the organizational level for balancing work and family, in terms of support of leader and co-workers, family-friendly policies and practices, and the complexity of work, have a significant impact on the work-family balance of an individual. A work-family balance makes an important contribution to work engagement, as this research showed that those who had a more balanced work and family life were more engaged in the workplace. In addition, work engagement is enhanced by the perception of the individual that the organization helps him or her to balance work and family life, regardless of how balanced his or her role in work and family life is.

## 4 Discussion

Data for Slovenia show that workers have difficulties in balancing work and family life [59]. Data from a survey conducted in Slovenia in 2016 by Kanjuo et al. [60] show that of the employees surveyed, 24% women and 30% men, have difficulties in work-family balance. An earlier study by Robnik [61] showed that problems in balancing work and family life mainly affect managers, employees with children, and employees with higher education, as well as employees who work directly with customers. According to the research data, they miss family activities or come home from work so emotionally exhausted that they cannot contribute to family life because of the time they spend with work commitments. 32.5% of respondents also felt that they do not spend enough time bringing up children, and just over half of employees do not spend enough time on leisure activities, according to their own estimates. In addition, survey data shows that workers also have difficulties in taking parental leave and other parental rights such as paternity leave, part-time leave, and childcare leave. The survey data also underlines the importance of work-family balance for workers, with 88% of respondents highlighting work-family balance as a very important or important aspect of work.

Practice in Slovenia thus shows that, despite relatively good legislation, organizations still do not sufficiently respect employees' needs for work-family balance and there are still violations of labor rights related to parenthood. This is partly due to the fact that employment-related legislation does not regulate the possibility of work-family balance, but treats

obligations arising from family life as a private matter for each individual. In fact, it is a matter of constantly adapting family life to the needs of the labor market, while there is not much dialog in the opposite direction [62].

At the same time as the problem of work-family balance, organizations face the problem of a lack of engagement at the workplace. Data from a Gallup survey shows that in Slovenia only 15% of employees are engaged, 70% are unengaged and 15% are actively unengaged [63].

On the basis of the above-mentioned results of previous research, we decided to review the current situation in the field of work-family balance in Slovenia and determine whether an improvement in work-family balance and various elements at the level of work-family balance could increase the work engagement of employeess. In this paper we examined the relationship between the support provided by the organization in balancing work and family life, work-family balance, and the work engagement of employees.

The first hypothesis was to determine whether factors at the level of the organization (and which ones) influence the balance between work and family. Numerous factors are highlighted in the literature. For the purposes of the study, we selected the most typical factors and those most frequently exposed in the literature to date [9,24,41,64] e.g. working hours, complexity of work, support from the leader, support from the co-workers, and help from the organization in the form of family-friendly policies and practices.

The results of the study showed a significant influence of all factors examined with the exception of working time, and thus we confirmed the first hypothesis that factors for work-family balance at the organizational level significantly influence the work-family balance of an individual. As in previous studies [11], our research confirmed that it is important for the individual to perceive the organization as family-friendly, as this has a significant impact on reducing work-family conflict and facilitating work-family balance.

The organization can have a significant impact on the work-family balance of employees through various factors, so the key question for organizations should be how to promote the improvement of the performance of employees in their individual roles and how to prevent conflicts between work and other life roles. Employers who support the work-family balance of employees benefit in several areas, such as reduced fluctuation and sick leave, as well as increased productivity, motivation, job satisfaction, and work commitment.

In the second hypothesis, we determined whether the work-family balance influences work engagement. The results of the study confirmed that the work-family balance has a positive effect on the work engagement of the individual. A higher level of work-family balance was associated with a higher level of work engagement. This can be explained by the fact that the perception of work-family balance is related to the individual's sense of support and value to the organization [32].

One of the possible reasons for the impact of work-family balance on work engagement is also the link between the two concepts. Research on the relationship between work and family involvement [65] has already shown that the relationship between work and family can have a significant impact on job and life satisfaction, and work and family involvement depends on this relationship. Similarly, another study [40] has shown that engagement in work and engagement in the family are positively related and beneficial to both. Thus, the positive emotional response that results from engagement in one role is intended to reinforce the individual's engagement in another role.

In the last hypothesis, we examined the impact of an individual's perception of an organization's support in work-family balance on work engagement. Although research investigating the impact of work-family balance on work engagement is rare, there are few studies that have examined the impact of an organization's support for work-family balance on work engagement [41–43,53]. The results of previous studies confirmed similar to the findings of our

research. That there is a connection between individual perceptions of organizational support for work-family balance and work engagement. The perception of work-family balance is related to the individual's feeling of being supported by the organization, so that he or she is more committed and engaged with the organization.

The results of the study confirm an important link between the organization's support for work-family balance, the individual's work-family balance, and work engagement. There is no doubt that the organization plays an important role in work-family balance, which was also confirmed by the results of the study. Work-family balance is important not only for the individual but also for the organization, as employees with a good work-family balance also show more engagement at work.

## 5 Conclusion

Theoretically, the work-family balance is becoming increasingly important, but practice shows that organizations have not succeeded in putting theory into practice because employees still have difficulties in balancing work and family life. The aim of organizations is to increase profits, often by overworking their employees. As a result, they have difficulty in balancing work and family life, and the trend towards burnout is also on the rise. Employers often don't notice that the time an employee has for oneself or one's family is extremely important not only for the individual, but in the long term also for the organiazation. The compatibility of work and family life has a considerable influence on the work engagement of employees.

This research provided several important findings that contribute to the current state of knowledge in the field of work-family balance and work engagement. The first is that organizational elements such as support of leader and co-workers, family-friendly policies and practices, and the complexity of work have a significant impact on the work-family balance of individuals. Second, work-family balance has a significant impact on the work engagement of employees. And the third finding is that an individual's perception of an organization's support for work-family balance has a significant impact on work engagement, regardless of the actual success of work-family balance.

We have only included some elements within the organization in this study, but have not considered some others that can also have a significant impact on the work-family balance, such as production processes or the working atmosphere. In future research it would also be useful to examine elements at the individual level, such as personality and demographic characteristics, as well as elements at the state level. Other aspects of private life, such as hobbies, culture, and leisure time, as well as the individual's ability to balance these with work, should also be investigated. The opportunities for future research are many and varied, since both work-family balance and work engagement are broad concepts that are influenced by many factors.

The practical results of this research implies that organizations should focus on work-family balance as an important aspect of work, improve and adapt work-family balance policies and practices to suit employees, and create a family-friendly climate in the organization—this would help to increase employee engagement and thus productivity. Many of the above elements can be systematically regulated by the state through legislation and the promotion of family-friendly entrepreneurship; not only declaratively, but also through legislation and facilitation for companies that ensure the successful balance of work and family life. At the same time, there is a need for effective and quality-oriented state control, which in practice leads to the implementation of family-friendly measures and the observance of standards in companies, and thus actually enables employees to successfully balance work and family life.

The profits of companies with excessive workloads are growing, but this process can only take place in the short term, so there is an urgent need to redefine the demands on workers

and working conditions. This is important for the workers, the organization, and for the state. The health care system cannot cope with the growing wave of diseases of people due to stress and burnout, there is also no officially prescribed treatment protocol, so that treatment is left almost entirely to the self-payment and initiative of the individual, which can push him or her to the social margins. First and foremost, however, everyone should be aware of the importance of work-family balance. The key question for organizations and the state should therefore be how to promote the improvement of the performance of employees in their individual roles and prevent conflicts between work and other life roles. Practices to balance work and family life create a "win-win situation" for both employees and the organization.

## Supporting information

**S1 Data.**
(XLSX)

**S1 File.**
(DOCX)

## Author Contributions

**Conceptualization:** Jasmina Žnidaršič, Mojca Bernik.

**Data curation:** Jasmina Žnidaršič.

**Formal analysis:** Jasmina Žnidaršič.

**Investigation:** Jasmina Žnidaršič, Mojca Bernik.

**Methodology:** Jasmina Žnidaršič.

**Project administration:** Mojca Bernik.

**Resources:** Jasmina Žnidaršič.

**Supervision:** Mojca Bernik.

**Writing – original draft:** Jasmina Žnidaršič.

**Writing – review & editing:** Jasmina Žnidaršič, Mojca Bernik.

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
