## [Decision Letter · Decision Letter 0]

27 Oct 2020

PONE-D-20-29272

Impact of work-family balance results on employee work engagement within the organization: The case of Slovenia

PLOS ONE

Dear Dr. Znidarsic,

Thank you for submitting your manuscript to PLOS ONE. After careful consideration, we feel that it has merit but does not fully meet PLOS ONE’s publication criteria as it currently stands. Therefore, we invite you to submit a revised version of the manuscript that addresses the points raised during the review process.

We look forward to receiving your revised manuscript.

Kind regards,

Grischa Beier

Academic Editor

PLOS ONE

Journal Requirements:

2.Please provide additional details regarding participant consent. In the ethics statement in the Methods and online submission information, please ensure that you have specified (1) whether consent was informed and (2) what type you obtained (for instance, written or verbal, and if verbal, how it was documented and witnessed). If your study included minors, state whether you obtained consent from parents or guardians. If the need for consent was waived by the ethics committee, please include this information.

4.We note that you have indicated that data from this study are available upon request. PLOS only allows data to be available upon request if there are legal or ethical restrictions on sharing data publicly. For more information on unacceptable data access restrictions, please see http://journals.plos.org/plosone/s/data-availability#loc-unacceptable-data-access-restrictions.

5. We note you have included a table to which you do not refer in the text of your manuscript. Please ensure that you refer to Table 1 in your text; if accepted, production will need this reference to link the reader to the Table.

Additional Editor Comments (if provided):

Dear Jasmina Znidarsic,

thank you very much for submitting your manuscript (number PONE-D-20-29272) to PLOS ONE.

In their comments, the reviewers recognize the importance of your line of research, and its potential to contribute to the scientific community's appreciation of the relationship between work-family balance and work engagement through the lens of organizational factors.

That being said, Reviewer 1 points to severe shortcomings in your study which you must address properly. Most importantly, you are asked to a) define your main dependent variables in a scientifically sound way, b) make clear how the sample scores on the main variables and therefore at least briefly present your basic analyses, c) be more precise and clear about some of the terms you use and d) better relate the statements in your results to the evidence you provide in your analysis. This last point of critique is also emphasized by Reviewer 2, who wants to see a stronger link between the results of your analysis and the conclusions you draw to your three hypotheses.

In short, based on this review, I feel that this article can be a very valuable contribution, but it will need to go through fundamental reworking. If you are able to do this in a resvision, your work will then indeed achieve its worthy goal of shedding light on the relationship between work-family balance and work engagement from in an organizational context.

Reviewers' comments:

Reviewer's Responses to Questions

**Comments to the Author**

1. Is the manuscript technically sound, and do the data support the conclusions?

Reviewer #1: Partly

Reviewer #2: Yes

2. Has the statistical analysis been performed appropriately and rigorously? 

Reviewer #1: I Don't Know

Reviewer #2: I Don't Know

3. Have the authors made all data underlying the findings in their manuscript fully available?

Reviewer #1: No

Reviewer #2: Yes

4. Is the manuscript presented in an intelligible fashion and written in standard English?

Reviewer #1: Yes

Reviewer #2: Yes

5. Review Comments to the Author

Reviewer #1: The authors are interested in the relationship between work-family balance (WFB) and work engagement (WE), through the lens of organizational factors that might affect both. This is an important question, and I am glad the authors are researching it.

Generally, the paper is a simplistic analysis of the relationship between work-family balance and work engagement, without any actual test of that relationship. Nevertheless, there is some benefit to this paper, and the data seem particularly interesting (despite the low response rate).

There are some fundamental issues that the authors need to address prior to publication, however. First, their definitions of their main dependent variables (DV), WFB and WE are missing entirely (although it is possible they intend for the reader to use the Andrew & Sofian definition of WE, it’s unclear). Though they do discuss how some other researchers have understood those terms, the reader is not clear how the authors do. Those definitions need to be clear. The variables also need to be clearly described in their Methods section. Currently, WE is not mentioned at all, and WFB has no examples of the types of measures that are included in their 4-item scale.

Second, the authors mention both WFB and work-life balance in various places in their paper. I am aware that some researchers have used these phrases interchangeably, but others have carefully denoted the difference. It would be good if the authors clarify their own position on this. Providing the definition will help.

Third, the authors make a point of saying at the beginning the importance of a gendered analysis in this substantive area, but they do not complete one in this paper. That is a glaring absence. Given their sample is balanced by sex, I would encourage them to minimally provide some basic analyses of overall scores on all of their independent and dependent variables by women & men. (I believe the other demographics can wait for a later paper.) Of course including gender in their regressions is ideal, along with interaction variables, but minimally creating some reader knowledge about gender differences is important.

This also speaks to a larger issue with the paper—the reader has no way to interpret the results on the DV because it is not clear how the sample scored on those variables. Generally, was the sample positive about organizational policies and practices? Leader support? Peer support? WFB or WE? Those basic analyses should be presented briefly so readers can ascertain the scale of the effects being discussed also.

Also in terms of methodology, I am unclear why they are calling this a case study. They are not looking within an organization, but rather across organizations. That’s a cross-sectional design. It would also benefit the reader if we knew if their sample demographics (including percent working in each sector) is comparable to population statistics. I also needed clarification between administration and management, because those terms are often used interchangeably in the literature.

In the Results, complexity of the work is said to be negatively associated with WFB, but the table shows positive. That needs clarification. Table 3 can also be more easily interpreted if we understand how WE is measured.

They also misspoke in the first and 2nd para after Table 3. H3 requires WE to be DV. If they did a multilevel regression with WFB as a mediating variable between organizational factors and WE, it should have been analyzed and presented as such. Table 4 does not look like that at all. So, the write-up and the table need to be better aligned to clarify the actual analysis conducted. This is especially important because on the next page the authors say: “A work-family balance makes an important contribution to work engagement, as research showed that those who had a more balanced work and family life were more engaged in the workplace.” The analyses shown do not verify that conclusion, as WFB and WE are not in the same analysis anywhere in the paper.

The last sentence prior to the conclusion also goes beyond their reported results by suggesting again a relationship between FWB and WE- in this case that the absence of FWB can be related to WE because of organizational factors. I see no evidence of this analysis in the paper.

Finally, there are some writing mistakes in the paper, such as misspellings or missing/extra words, but the paper is generally well-written.

Reviewer #2: Very interesting paper, but I have some suggestions for your considerations:

1) the key words should be revised, "Work-family balance" and "work engegement" should be fine, but "model" maybe is not a good term here, you can consider some other words, like "organization management"?

2) More details should be added for proposing the three hypotheses, and it should be taken as an idenpendent section.

3) I suggest to divide the section 3 into 3 sub-sections by corresponding to the three thypotheses.

6. PLOS authors have the option to publish the peer review history of their article (what does this mean?). If published, this will include your full peer review and any attached files.

Reviewer #1: No

Reviewer #2: **Yes: **Bing Xue

---

## [Author Response · Author response to Decision Letter 0]

10 Nov 2020

Dear editors and reviewers,

The authors would like to thank the editor and the reviewers for their time and for considering our manuscript for publishing in PLOS ONE. 

We have edited the manuscript to address reviewers' concerns and comments. The authors believe that the whole paper has been improved and we are happy to forward it back to you for your consideration and publication. 

The authors' revisions are presented in the Rebuttal letter with a point-by-point response to the comments. We are also sending a revised manuscript with track changes (TC) and a clean version of the revised manuscript (without TC) as well as revised figure and document with means, meadians and SD of all variables.

We hope that the reviewers' comments were adequately addressed.

Best wishes,

Authors

---

## [Decision Letter · Decision Letter 1]

2 Dec 2020

PONE-D-20-29272R1

Impact of work-family balance results on employee work engagement within the organization: The case of Slovenia

PLOS ONE

Dear Dr. Znidarsic,

Thank you for submitting your manuscript to PLOS ONE. After careful consideration, we feel that it has merit but does not fully meet PLOS ONE’s publication criteria as it currently stands. Therefore, we invite you to submit a revised version of the manuscript that addresses the points raised during the review process.

We look forward to receiving your revised manuscript.

Kind regards,

Grischa Beier

Academic Editor

PLOS ONE

Additional Editor Comments (if provided):

Dear authors,

thank you for carefully and thoroughly revising your manuscript according to the comments received in the first round of review. This has substantially improved the quality of the manuscript.

However, there are still some minor issues, which will require your attention. Please pay special attention to the question of gender differences in job complexity, raised by reviewer 1. You should also follow the advice to use a more gender neutral language in the sections that have been added previously.

Please also be aware, that your manuscript will also require careful language editing through the authors prior to publication.

Reviewers' comments:

Reviewer's Responses to Questions

**Comments to the Author**

1. If the authors have adequately addressed your comments raised in a previous round of review and you feel that this manuscript is now acceptable for publication, you may indicate that here to bypass the “Comments to the Author” section, enter your conflict of interest statement in the “Confidential to Editor” section, and submit your "Accept" recommendation.

Reviewer #1: All comments have been addressed

Reviewer #2: All comments have been addressed

2. Is the manuscript technically sound, and do the data support the conclusions?

Reviewer #1: Yes

Reviewer #2: Yes

3. Has the statistical analysis been performed appropriately and rigorously? 

Reviewer #1: I Don't Know

Reviewer #2: Yes

4. Have the authors made all data underlying the findings in their manuscript fully available?

Reviewer #1: Yes

Reviewer #2: Yes

5. Is the manuscript presented in an intelligible fashion and written in standard English?

Reviewer #1: No

Reviewer #2: Yes

6. Review Comments to the Author

Reviewer #1: The authors have substantially rewritten their paper, clarifying much of their terminology and analyses. Their results are now clear. The conclusions they reached in their prior paper have now been largely confirmed here. There is a bit of redundancy added into the literature review now, but it does not substantially detract from the paper's arguments.

Their writing, however, continues to have problems and will require careful editing prior to publication. These mistakes are minimal, but frequent.

One mistake remains in the paper- on p 19 the authors state that men's jobs are more complex. However, they reverse the codes for complexity to make this argument (found 3 other places in the document coded differently). I find it ironic that they invoked a masculine understanding of complexity to give men more complex jobs (by misinterpreting their own coding) when their own data suggest that's not the case.

Finally, in the new sections added the authors have used solely male pronouns. Given that the rest of the document employes a more gender neutral approach, I recommend they continue with that neutral approach and revise the new sections accordingly.

Reviewer #2: I would like to suggest that this manuscript should be considered for publication in the journal PLOS one

7. PLOS authors have the option to publish the peer review history of their article (what does this mean?). If published, this will include your full peer review and any attached files.

Reviewer #1: No

Reviewer #2: **Yes: **Bing Xue

---

## [Author Response · Author response to Decision Letter 1]

17 Dec 2020

Dear editors,

We would like to thank the editor and the reviewers for their time and for considering our manuscript for publishing in PLOS ONE. 

We have edited the manuscript to address reviewers' concerns and comments. We believe that the whole paper has been improved and we are happy to forward it back to you for your consideration and publication. 

The authors' revisions are presented in the Rebuttal letter with a point-by-point response to the comments.The article was edited by an official English proofreader.

We hope that the reviewers' comments were adequately addressed.

Best wishes,

Authors

Dear Reviewer 1,

We would like to thank the reviewer for the comment. We reviewed and edited the article. We have corrected the data on p19. We thank the reviewers for a detailed review and exposure of nonconformities. The correction can be seen from the text. We corrected solely male pronouns where it was noticeable and didn’t make sense given the context. The article was also edited by an official english proofreader.

We hope that the reviewers' comments were adequately addressed.

Best wishes,

Authors

Dear Reviewer 2,

We would like to thank the reviewer for the comment.

Best wishes,

Authors

---

## [Editor Report · Decision Letter 2]

22 Dec 2020

Impact of work-family balance results on employee work engagement within the organization: The case of Slovenia

PONE-D-20-29272R2

Dear Dr. Znidarsic,

We’re pleased to inform you that your manuscript has been judged scientifically suitable for publication and will be formally accepted for publication once it meets all outstanding technical requirements.

Kind regards,

Grischa Beier

Academic Editor

PLOS ONE
---

## [Editor Report · Acceptance letter]

11 Jan 2021

PONE-D-20-29272R2 

Impact of work-family balance results on employee work engagement within the organization: The case of Slovenia 

Dear Dr. Žnidaršič:

I'm pleased to inform you that your manuscript has been deemed suitable for publication in PLOS ONE. Congratulations! Your manuscript is now with our production department. 

Kind regards, 

on behalf of

Dr. Grischa Beier 

Academic Editor

PLOS ONE